# Machine-Learning-Based Methodology for Estimation of Shoulder Load in Wheelchair-Related Activities Using Wearables

**DOI:** 10.3390/s23031577

**Published:** 2023-02-01

**Authors:** Sabrina Amrein, Charlotte Werner, Ursina Arnet, Wiebe H. K. de Vries

**Affiliations:** 1Rehabilitation Engineering Laboratory, Department of Health Science and Technology, ETH Zurich, 8049 Zurich, Switzerland; 2Swiss Paraplegic Research, Guido A. Zächstrasse 4, 6207 Nottwil, Switzerland; 3Spinal Cord Injury Center, University Hospital Balgrist, 8008 Zurich, Switzerland

**Keywords:** machine learning, wearable sensors, IMU, EMG, biomechanical modeling, spinal cord injury, joint force, shoulder

## Abstract

There is a high prevalence of shoulder problems in manual wheelchair users (MWUs) with a spinal cord injury. How shoulder load relates to shoulder problems remains unclear. This study aimed to develop a machine-learning-based methodology to estimate the shoulder load in wheelchair-related activities of daily living using wearable sensors. Ten able-bodied participants equipped with five inertial measurement units (IMU) on their thorax, right arm, and wheelchair performed activities exemplary of daily life of MWUs. Electromyography (EMG) was recorded from the long head of the biceps and medial part of the deltoid. A neural network was trained to predict the shoulder load based on IMU and EMG data. Different cross-validation strategies, sensor setups, and model architectures were examined. The predicted shoulder load was compared to the shoulder load determined with musculoskeletal modeling. A subject-specific biLSTM model trained on a sparse sensor setup yielded the most promising results (mean correlation coefficient = 0.74 ± 0.14, relative root-mean-squared error = 8.93% ± 2.49%). The shoulder-load profiles had a mean similarity of 0.84 ± 0.10 over all activities. This study demonstrates the feasibility of using wearable sensors and neural networks to estimate the shoulder load in wheelchair-related activities of daily living.

## 1. Introduction

Manual wheelchair users (MWUs) highly rely on their upper limbs for independent mobility during daily life. The shoulder complex becomes the main source of power for locomotion and activities of daily living (ADL). The high prevalence of shoulder problems reported in MWUs (30–70%), such as pain, pathologies, and a limited range of motion [1,2,3,4,5], drastically limits MWUs’ participation in daily life, reduces their quality of life, and ultimately increases the health care cost for society [2].

Some ADL are known for an increased shoulder load, such as wheelchair (WC) propulsion, weight-relief lifts, and transfers [6,7]. These tasks are performed multiple times per day, which results in a high exposure of the shoulder [6,8]. Further, the risk factors for joint overload are known, namely the magnitude, frequency, and duration of joint load. If one or a combination of these three factors becomes too high, they can cause shoulder joint overload and consequently shoulder problems [9,10]. A methodology for shoulder-load estimation in MWUs in daily conditions is therefore of substantial importance from two viewpoints: (1) to monitor an MWU’s daily shoulder load and (2) to study the longitudinal effects of shoulder load with respect to shoulder pain and pathologies. Knowledge of the daily shoulder load is imperative for the implementation of potential interventions to reduce the shoulder load and shoulder pain during ADL.

Until now, only laboratory-based measurements have examined the shoulder load during WC-related tasks [11,12,13]. Here, musculoskeletal modeling is used to estimate the joint load, a method widely accepted as the gold standard. Musculoskeletal modeling requires kinematic data and kinetic data as input to calculate the joint load. Kinetic data alone, acquired, for example, using force plates or knowledge of handled weights, but especially the combination of kinematic and kinetic data can only be reliably measured within a laboratory setting [14,15]. Hence, musculoskeletal modeling for joint-load calculation is restricted to a laboratory setting. As with most laboratory measurements, the extensive equipment as well as the restricted area covered by the motion capture cameras used for kinematic data acquisition may hinder the participants’ natural movements or execution of daily tasks. Furthermore, these measurements only represent a snippet of the daily shoulder loading in manual WC users. Additionally, musculoskeletal modeling based on laboratory data allows assessing only one of the three risk factors known from the literature for joint overload, namely the magnitude. To fully characterize shoulder-loading activities, not only the magnitude but also the frequency and duration of specific ADL need to be considered.

These drawbacks of musculoskeletal modeling underline the importance of a new, non-laboratory-based methodology for joint-load estimation in daily conditions. The use of alternative methods, such as wearable sensors, has been excessively investigated in biomechanical applications to address some of the limitations biomechanical laboratories possess [16,17,18,19]. Wearable sensors such as inertial measurement units (IMU) and electromyography (EMG) sensors do not require a complex laboratory setup but can track movement or muscle activity in daily conditions. Wearable sensors can easily be attached with straps to the participant or the equipment. They are small, lightweight, and wireless, thus unobtrusive measurement tools allowing full movements. First promising results by Goodwin et al. demonstrated the usefulness of wearable sensors in characterizing humeral elevation, a possible risk factor for shoulder pathology in MWUs with a spinal cord injury [16]. 

To evaluate sensor data, especially in a sparse measurement setup, machine learning techniques are necessary [20,21]. In recent years, IMU and machine learning techniques, including artificial neural networks (ANN), have been successfully applied in both classification and prediction tasks in biomechanical settings [21,22,23,24,25,26,27]. Traditional machine learning algorithms such as Random Forest, Support Vector Machine, etc., require time-intensive feature engineering and manual feature extraction. This is a serious drawback when handling complex, dynamic data with time dependencies, such as human movement and force exertions [21,28]. The advantage of ANNs is their ability to realize an arbitrary mapping of one vector space onto another vector space. ANNs capture some previously unknown information hidden in given data (training data set), learn from it, and apply the learned input–output relationship to new data. This allows a prediction of the behavior of a system with respect to unseen data [23,29]. 

Recurrent neural networks, including long short-term memory (LSTM) networks, use time dependencies in data to make predictions. By updating an internal state, previous information is retained and will affect future predictions, while irrelevant information is disregarded [30]. The key difference with ANN is that bidirectional recurrent neural networks, including bidirectional LSTM (biLSTM), propagate the input in two layers. In a forward layer, the input passes from past to future, while in a backward layer, the input passes from future to past. In this way, not only previous information but also future information will affect the current prediction [31,32,33]. This unique characteristic of biLSTM appears to be a huge advantage in the prediction of joint load from wearable sensor data. LSTM networks have been applied for the estimation of lower-limb kinematics [34,35] and the estimation of ground reaction forces and knee-joint kinetics [23] from IMU data. However, LSTM networks are intensive to train: large data sets and considerable memory capacity are required, and the training is time-consuming [30]. These are the main disadvantages of this method. Linear neural networks, on the other hand, are easy to train and often more generalizable if only small data sets can be provided [36]. Stetter et al. utilized wearable sensors in combination with a linear neural network for knee joint force estimation during sports movements [25]. Similar work has been conducted on upper extremities by de Vries et al. [37]. They published a proof of concept for shoulder joint force estimation, demonstrating the feasibility of utilizing IMU and EMG sensors together with a linear neural network for the estimation of shoulder joint force. This project builds upon the work of de Vries et al. [37] by further exploring the feasibility of utilizing machine learning techniques for shoulder-load estimation in WC-related ADL. So far, no non-laboratory-based methodology to assess shoulder load during ADL in MWUs is available.

Therefore, this project aims to develop and evaluate a non-laboratory-based methodology for the estimation of shoulder joint load in daily conditions in WC-related ADL using wearable sensor data and ANNs.

## 2. Materials and Methods

### 2.1. Data Collection

Ten able-bodied participants (7 female; age 39 ± 9.4 years; height 169 ± 9.1 cm; weight 66 ± 12 kg) with no pain in the upper extremities and experienced in WC-related activities participated in this study. The study was approved by the Ethikkommission Nordwest- und Zentralschweiz (EKNZ, Project-ID: 2020-01961). The study follows the ICH Good Clinical Practice Guidelines and the Swiss regulation on research involving human beings. All participants were informed of the experimental procedures and gave informed written consent prior to the measurements. Before the actual measurements, participants were familiarized with the WC-related activities until they felt comfortable executing the activities. 

The participants were equipped with IMU sensors, EMG sensors, motion capture marker clusters, and a Smartwheel as shown in Figure 1. Five IMU sensors (Shimmer3 IMU Unit, Shimmer, Dublin, Ireland) (sampling frequency 128 Hz, ±8 g accelerometer, ± 2000°/s gyroscope) were attached to the participants’ lower right arm, upper right arm, and thorax, and on the WC frame and wheel. IMU sensors collect information about the acceleration and the angular velocity of the body segment the sensor is attached to.

EMG data (Shimmer3 EMG Unit, Shimmer, Dublin, Ireland) (1024 Hz, gain 12) were collected from the medial deltoid and the long head of the biceps muscle of the participants’ right arm. An eight-camera marker-based motion capture system (Oqus, Qualisys AB, Gothenburg, Sweden) (100 Hz) was used to obtain upper body kinematics conforming to Wu et al. [38]. A SmartWheel (Three Rivers Holdings LLC, Mesa, Arizona, USA) (240 Hz) for collecting propulsion kinetics replaced the original right wheel of a standard active wheelchair (Küschall Compact 2017, Küschall AG, Witterswil, Switzerland). The data-collecting systems were synchronized during post-processing by cross-correlation. The participants executed a well-recognizable motion each time data acquisition was initiated. 

The participants performed six different activities exemplary of the daily living of MWUs. All participants performed the activities in the same order, namely a first weight-relief lift of 10 s duration; a specified sequence of WC propulsion maneuvers in a restricted space; WC propulsion at 0.56 m/s and 1.1 m/s at 0% inclination, and 0.56 m/s at 6% inclination for 30 s each; a second weight-relief lift of 10 s duration; ascending and descending a short ramp of 12% inclination; manual material handling, specifically placing a weight of 2 kg on three different levels of a shelf, followed by putting the weight into the back pocket of the WC; and desk work for 30 s, such as typing, working with the computer mouse, and making a phone call. Each of these activities was recorded once for each participant.

### 2.2. Data Processing and Biomechanical Modeling

The kinematic data from the motion capture system were filtered (Butterworth low-pass filter, fourth order, cut-off frequency 6 Hz). The SmartWheel data were offset corrected, filtered (Butterworth low-pass filter, fourth order, cut-off frequency 20 Hz), and downsampled to 100 Hz. Through the recorded upper-body kinematics and external forces (SmartWheel data for WC propulsion, weight-relief lifts, and ascending/descending a ramp; known weight of 2 kg for manual material handling), 3D shoulder-joint reaction forces (SJRF) were estimated using musculoskeletal modeling within OpenSim [39]. Each participant was individually modeled with a validated upper-extremity model scaled to the participant’s height and weight [40]. During static optimization within the OpenSim processing pipeline, the data were downsampled to 25 Hz to reduce the computation time. The equation solved during static optimization considers constraints on the glenohumeral joint force direction which ensures that the calculated muscle forces produce sufficient stabilizing glenohumeral joint compression. The resultant SJRF was filtered (Butterworth low-pass filter, fourth order, cut-off frequency 4 Hz). 

The IMU signals (acceleration and angular velocity) were filtered (Butterworth low-pass filter, fourth order, cut-off frequency 10 Hz) and downsampled to 25 Hz to correspond to the specific SJRF signals. EMG data were high-pass filtered (Butterworth, fourth order, cut-off frequency 20 Hz), corrected for offset, rectified, low-pass filtered (Butterworth, fourth order, cut-off frequency 3 Hz), normalized using submaximal isometric contraction, and downsampled to 25 Hz. The participants executed two static postures for normalization of the EMG signal by submaximal isometric contraction. For normalization of the medial deltoid, the participants were holding a weight of 2 kg in 90° abduction with the elbow extended and the thumb pointing frontally. For normalization of the long head of the biceps, the participants were holding a weight of 2 kg in elbow flexion with the forearm pointing frontally and the thumb pointing upwards.

### 2.3. Neural Network Modeling

The ANN developed for this study maps the signals from the 5 IMUs (tri-axial acceleration, tri-axial gyroscope) and 2 EMG sensors to the SJRF time series from the musculoskeletal modeling. The sensor signal matrix (N × 32, where N depicts the trial length) served as input and the SJRF matrix (N × 3) served as target. Input and target were standardized by removing the mean and scaling to unit variance. Standardization happened independently on each input signal by computing the relevant statistics on the training data. A biLSTM was chosen as the preferred model due to its before-mentioned characteristics. The ANN was set up with the PyTorch library in Python (version 3.9.7). The biLSTM model consisted of three biLSTM layers [31,41]. Each biLSTM layer was followed by a dropout layer with a dropout probability of 0.37 to reduce overfitting. The three biLSTM layers each contained 128 neurons and were followed by a ReLU (Rectified Linear Unit) activation function. The model was trained for a maximum of 200 epochs. Training stopped if the validation loss did not decrease for six consecutive epochs. The model parameters resulting in the lowest validation loss during training were saved and reloaded for evaluation on the test data. During the initialization of an ANN, random weights are assigned to all internal connections, followed by the training process. For each initialization, this random weight assignment might result in a different outcome of the training process and hence in a different performance of the trained ANN [42]. Therefore, the model was initialized, trained, and evaluated for ten iterations. The model was tested on unseen data using a leave-one-out validation procedure. Figure 2 shows an overview of the neural network modeling pipeline.

Different cross-validation strategies, different sensor setups, and different model architectures were examined. While it would have been interesting to train every cross-validation strategy with each sensor setup and model architecture, a stepwise approach was followed to systematize the data for evaluation and to reduce the processing time. This stepwise approach led to the potentially best combination of cross-validation strategy, sensor setup, and model architecture.

Cross-Validation Strategy:

In step one, two different cross-validation strategies were compared. First, the biLSTM model was evaluated using a generalizable leave-one-subject-out (LOSO) cross-validation strategy. Specifically, the neural network was trained on all data from all but one participant (training set) and then tested on the data of the remaining participant (test set). This generalizable cross-validation strategy was compared to a subject-specific strategy, where the biLSTM was evaluated using the leave-one-trial-out (LOTO) strategy. Here, the training set consisted of all but one subtrials of one participant and the test set consisted of the remaining subtrial of the same participant.

To prepare the data for the LOTO train-test split, the activities were divided into two, three, or six subtrials, depending on the activity’s length and characteristics. Specifically, this resulted in six subtrials of WC propulsion on a treadmill (two for each condition), three subtrials of WC propulsion in restricted space, one subtrial each of ascending and descending a short ramp, two subtrials of weight-relief lift, three subtrials of manual material handling, and three subtrials of desk work. Seven unrelated subtrials from four different participants had to be excluded from further analysis due to a fault in the muscles’ wrapping paths within the OpenSim processing pipeline, resulting in a total of 183 subtrials (69,971 samples) (10 participants × 19 subtrials—7 invalid subtrials).

The cross-validation strategy leading to better results was specified as the cross-validation strategy for the next steps in the analysis.

Sensor Setup:

In step two, the input matrix was reduced towards a more pragmatic approach. For this pragmatic approach, the sensor setup was reduced to a sparse setup. Only the data from the IMUs attached to the participant’s upper arm, the data from the two EMG sensors, and the data from both IMUs attached to the WC contributed to the input matrix (N × 20). The IMU attached to the participant’s upper arm was included because it is the IMU placed on the segment distal to the shoulder, the joint of interest. Furthermore, upper-arm movement presumably provides more diverse information for the different ADL than the trunk movement. EMG was regarded as necessary information for SJRF estimation, as it delivers information on the musculoskeletal response of the activity and is related to the exerted force. The sensor setup leading to better results was specified as the sensor setup for the last step of the analysis.

Model Architecture:

In step three, the model architecture was simplified to minimize the processing time. The complex biLSTM model was compared to the simpler linear model. The linear model had two hidden layers, one with 250 and one with 100 neurons. A ReLU activation function followed each hidden layer. Similar to the training procedure of the biLSTM model, the linear model was trained for a maximum of 200 epochs or until the validation loss did not decrease for six consecutive epochs. The model parameters resulting in the lowest validation loss during training were saved and reloaded for evaluation on the test data. The linear model was initialized, trained, and evaluated for ten iterations.

### 2.4. Statistical Analysis

The total SJRF ($F_{tot}$) was calculated as the Euclidean norm of the three individual components ($F_{x},\;F_{y},\;F_{z}$). The similarity between the ground-truth $F_{tot}$ from the musculoskeletal model and the ANN-predicted $F_{tot}$ was analyzed using Pearson’s correlation coefficient (PCC) and relative root-mean-squared error (rRMSE), which normalizes the RMSE by the range of the ground-truth $F_{tot}$. For evaluation of the neural networks’ performances, the subtrials were concatenated to the original eight activities. The neural networks’ performances were analyzed on an activity level and a participant level. For analysis on the activity level, each participant’s mean performance across the ten iterations for the specific activity was calculated. As there were ten different participants, this led to a total of ten PCC and rRMSE values for each activity, one for each participant. For analysis on the participant level, all eight activities of one iteration were concatenated for each participant. Concatenation of all eight activities formed a complete iteration. Every complete iteration was analyzed, resulting in ten PCC and rRMSE values for each participant, one for each complete iteration.

The shoulder-load profiles were regarded as histograms and evaluated as such. To determine the similarity between the predicted shoulder-load profile ($\hat{y}$) and the ground-truth shoulder-load profile (y), the intersection (I) of the two histograms was calculated according to Swain and Ballard’s [43] conform Equation (1):(1)$$I\left(y,\;\hat{y}\right)=\frac{\sum_{i=1}^{n}\min\left(y_{i},\;{\hat{y}}_{i}\right)}{\sum_{i=1}^{n}y_{i}}$$

## 3. Results

### 3.1. Cross-Validation Strategy

In step one, the two cross-validation strategies (LOTO vs. LOSO) were compared. Figure 3A,B shows the comparison between the subject-specific LOTO organization strategy and the generalizable LOSO organization strategy when using the biLSTM model. In Figure 3A, PCC and rRMSE are depicted for the ANN’s performance on the participant level, while 3B shows the same metrics for the ANN’s performance on the activity level. Using the subject-specific LOTO validation strategy results in a distinctly better prediction accuracy (higher PCC values, lower rRMSE values) for all participants but participants 3, 4, and 5. The mean prediction accuracy on the participant level for the LOTO validation strategy was PCC = 0.78, rRMSE = 8.32%, while the mean prediction accuracy for the LOSO validation strategy was PCC = 0.75, rRMSE = 9.89%. Hence, the LOTO was chosen as the preferred validation strategy. 

Figure 3B shows a large difference in the prediction accuracy between participants for the weight-relief lift activity when using the LOTO organization strategy but not for the LOSO organization strategy.

### 3.2. Sensor Setup

In step two, the two different sensor setups (complete vs. sparse) were compared. Figure 4A,B visualizes the biLSTM model’s performances on the participant level and the activity level, respectively, when combining the LOTO cross-validation strategy with different sensor setups. Figure 4A shows that the model performs better having the complete sensor data available on the participant level (PCC = 0.78, rRMSE = 8.32%). With the sparse sensor setup, the mean prediction accuracy on the participant level was PCC = 0.74, rRMSE = 8.93%. However, on the activity level, these differences were negligible, as shown in Figure 4B. Due to the small differences between the two sensor setups the sparse setup was chosen.

### 3.3. Model Architecture

In step three, the model architecture was simplified to minimize the processing time. Figure 5A,B compares the complex biLSTM model with the simpler linear model when using the LOTO cross-validation strategy and the sparse sensor setup. Using the linear model, the mean prediction accuracy on the participant level was PCC = 0.65, rRMSE = 10.30%. Overall, the biLSTM outperforms the linear model on both the activity and the participant level (PCC = 0.74, rRMSE = 8.93%).

### 3.4. Final Model

Figure 6 provides an overview of the stepwise approach. It shows the setups used for the individual steps, the results of the individual steps, and the setup of the final model. Specifically, the final model combines the subject-specific LOTO cross-validation strategy with the biLSTM network and the sparse sensor setup.

The results of the final model show inter-participant differences (Figure 5A). Likewise, the prediction accuracy on the activity level follows a distinct pattern, independent of the model architecture (Figure 5B). Concretely, WC propulsion in a restricted space achieves a lower prediction accuracy than manual material handling, which in turn has a lower prediction accuracy than the three WC propulsion activities on the treadmill.

Figure 5B shows that the three WC propulsion activities on the treadmill reach the highest prediction accuracy with a small difference between participants.

Figure 7 shows the predicted $F_{tot}$ and the ground-truth $F_{tot}$ of one complete iteration for one participant using the final model as described in Figure 6.

### 3.5. Shoulder-Load Profiles

Figure 8 shows the shoulder-load profiles for all ten participants and a random iteration when using the final model as described in Figure 6. The mean intersection values are high for all participants with a consistently low standard deviation (I ≥ 0.81 ± 0.01), as listed in Table 1.

Figure 9 shows the shoulder-load profiles for all activities of one participant exemplary of all participants. Weight-relief lift has the lowest mean intersection value and a distinctly higher standard deviation than all other activities, as listed in Table 2.

## 4. Discussion

This study investigated the usage of ANNs for the continuous estimation of SJRF from wearable sensors in WC-related ADL. For this reason, different cross-validation strategies, different sensor setups, and different model architectures were examined.

### 4.1. Cross-Validation Strategy

The results indicate that a subject-specific cross-validation strategy (LOTO) attains a higher prediction accuracy than a generalizable cross-validation strategy (LOSO). A generalizable strategy would have been preferable, as such a model needs to be trained only once on a broad spectrum of the activities of interest and can then reliably predict data from unseen participants. A model based on a subject-specific strategy, on the other hand, needs to be trained anew for each participant. Such a model is more resource-intensive, as it requires the collection of subject-specific data within a laboratory setting and time-consuming training of the model.

The large difference between participants observed for the weight-relief lift activity using the LOTO organization strategy can be explained by investigating the outlier, participant 5, more closely. For this participant, the LOSO cross-validation strategy performed clearly better than the LOTO strategy. Closer investigation of that participant’s data revealed that one weight-relief lift had to be excluded from the data set due to an error within the OpenSim processing pipeline. Only one weight-relief lift activity remained in this participant’s data set. When this weight-relief lift constituted the test set, no similar data were included in the training set. Accordingly, following the LOTO approach, the prediction of this specific activity was poor, as shown in Figure 10A. In contrast, when a participant’s data set contained two weight-relief lift activities, one was always included in the training set if the other constituted the test set, as visualized in Figure 10B.

Using the generalizable LOSO organization strategy in this case massively increases the prediction accuracy. The model has seen weight-relief lift activities from other participants during training before predicting the data of participant 5. This observation is in line with the previously published literature, which all propose to focus the training data on the type of activity expected to be encountered in the test data [30,35,37].

### 4.2. Sensor Setup

For comparison of the two sensor setups, the LOTO cross-validation strategy and the biLSTM model were used. The ANN using the complete sensor setup performed only slightly better than the ANN using the sparse sensor setup. The IMU sensors attached to the participant’s thorax and lower right arm, which are ignored for the sparse setup, possibly provide mostly redundant information already provided by the sensor attached to the participant’s upper right arm. This finding is relevant insofar as it increases the applicability of the methodology. Using only one body-bound sensor is more convenient and less restrictive for the participant. Hence, the measured activities will be closer to the participant’s natural activities. Additionally, a sparse senor setup is less resource-intensive. Preparing the participant, processing the data, and training the ANN will demand less time. The benefits of the sparse sensor setup outweigh the small accuracy tradeoff and the sparse setup is therefore the preferred option to the complete setup. The usage of an even sparser sensor setup was not investigated. Reducing the setup further is not expected to improve the quality of the data. The WC-bound sensors do not hamper the participant’s convenience or restrict the movements in any case.

### 4.3. Model Architecture

For comparison of the two model architectures, the LOTO cross-validation strategy and the sparse sensor setup were used. The results suggest that the linear model cannot learn the intricate relation between sensor data and SJRF.

### 4.4. Final Model

With each iteration, the ANN is initiated and random weights are assigned. The small difference between iterations implies that the ANN’s predictive power is independent of the random weights’ assignment. This suggests that for future application in research it is sufficient to initialize and train a single ANN.

The distinct pattern observed for participants and activities indicates that the prediction accuracy for an individual participant and activity is highly dependent on the training data. The absolute accuracy of the predicted SJRF changes with the type of ANN used. However, the prediction accuracy in relation to other participants or activities is consistent, independent of the ANN used.

The prediction accuracy of the ANN strongly varied between activities. Here, the accuracy seems to correlate with the duration of the initial activity. Long activities were split into several long subtrials. That way, more data focused on the activity of interest were included in the training set, which improved the performance on the test set. Ascending and descending a ramp is a short activity. When split into subtrials, only a little similar data were included in the training set. This is reflected in the low prediction accuracy. These findings further underline the importance of including sufficient data in the training set. Sufficient data is difficult to define and a topic for further research. For this study, including a full activity and not only subtrials in the training set could potentially already improve the prediction accuracy. 

A potential reason for the high prediction accuracy observed in the three WC propulsion activities on the treadmill is the repeatable characteristics of the activity. In contrast, activities with a higher rate of variation such as WC propulsion in a restricted space or manual material handling reach poorer accuracies. Stetter et al. made a similar observation for the prediction of knee joint forces, where they observed the highest predictive power for moderate running and only limited predictive power for activities with higher variation, such as sprint starts and full stops [25].

Stetter et al. [25] used an ANN for the prediction of the three individual components of knee joint force ($F_{x},\;F_{y},\;F_{z}$). The ANN-predicted knee joint forces yielded PCC values ranging from 0.25 to 0.94 and rRMSE values ranging from 14.2% to 45.9%, depending on the component and the activity. The PCC value for the total knee joint force, although not reported, seems to be similar to our results; the rRMSE value is lower in our study. De Vries et al. [37] trained a linear model to predict SJRF during ADL in one healthy subject based on wearable sensors. They reported a good to excellent prediction accuracy (intraclass correlation coefficients ranging from 0.83 to 0.98). The use of different evaluation metrics makes a direct comparison of the results between our study and the study of de Vries et al. difficult. Additionally, the set of activities measured is more diverse and complex in our study. 

### 4.5. Shoulder-Load Profiles

There is a consistently high similarity between the ground-truth and the predicted shoulder-load profiles for all participants and most activities. One possible explanation for this high similarity between the shoulder-load profiles is the effect of binning. While a small deviation from the true SJRF has a possibly high effect on the prediction accuracy of SJRF, binning nullifies this effect. Another possible explanation is the equalization of deviations over time. The SJRF is predicted too high for some activities and too low for others. Both false predictions have a decisive effect on the predicted SJRF and hence on PCC. With shoulder-load profiles, however, these false predictions will equalize over time.

A potential reason for the noticeably low intersection value of the weight-relief lift has been extensively discussed in Section 4.1.

### 4.6. Limitations and Future Research

Caution is required as the methodology was developed using data from able-bodied participants. A validation study with spinal-cord-injured participants is in progress. The results show that a subject-specific algorithm exceeds a generalizable algorithm in prediction accuracy. From a pragmatic point of view, a generalizable algorithm would be preferable to a subject-specific algorithm as it is less resource-intensive. Future research could focus on improving the prediction accuracy of the generalizable algorithm either by increasing the data set and hence the amount of training data or by performing a sensor-to-segment alignment and hence reducing the variability within the training data. The ground-truth SJRF for this study was based on musculoskeletal modeling. Musculoskeletal modeling in turn is based on several assumptions, such as intrinsic muscle parameters that cannot be measured in vivo, and therefore has its specific limitations. Providing the best means of reference data for the ANN modeling could help to predict the SJRF more precisely. A further limitation of the work might be seen in the absence of a wide comparison of available machine learning methods. This absence is justified by the careful selection of methods that have previously been successfully applied to similar problems in the estimation of joint load from wearable sensor data.

## 5. Conclusions

This work is a considerable step towards assessing shoulder load in daily life, which has not been achieved yet. The results of this study prove the feasibility of utilizing neural networks for quantifying the shoulder load in WC-related ADL based on IMU and EMG data. Specifically, the shoulder-load profiles for participants showed exceptional agreement between the ground-truth SJRF and the ANN-predicted SJRF. Knowledge of shoulder-load profiles combined with knowledge of the type of activity performed will introduce relevant targets for the reduction in shoulder-joint load, and hence reduction in shoulder pain and shoulder pathologies.

## Figures and Tables

**Figure 1 sensors-23-01577-f001:**
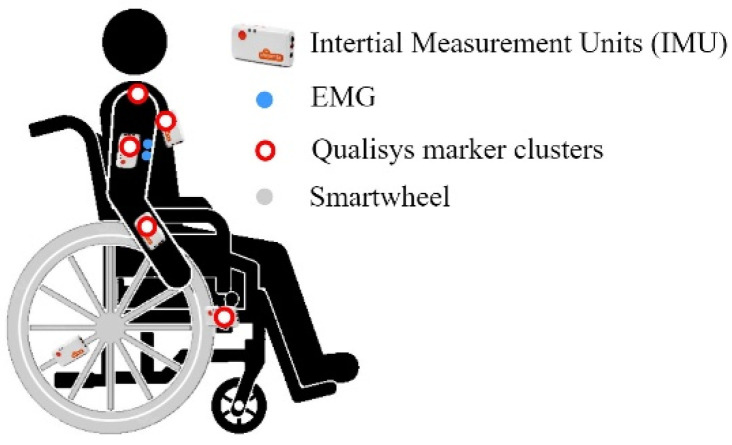
Equipment of a participant.

**Figure 2 sensors-23-01577-f002:**
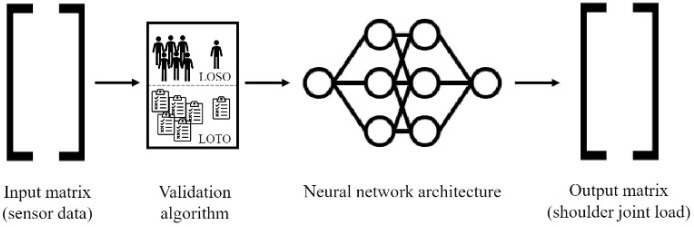
Overview of the neural network modeling pipeline.

**Figure 3 sensors-23-01577-f003:**
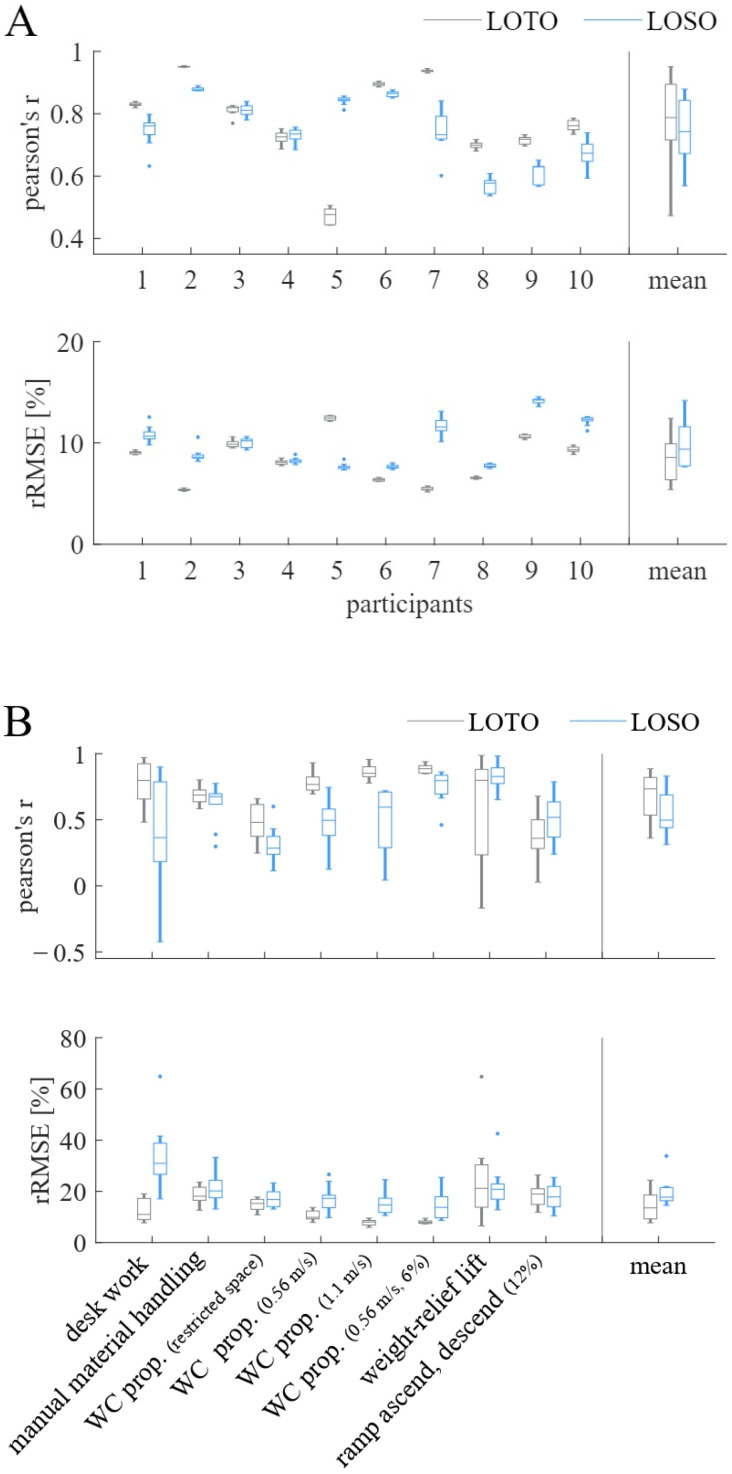
Pearson’s r and relative root-mean-squared error (rRMSE) for comparison of leave-one-trial-out (LOTO) and leave-one-subject-out (LOSO) cross-validation strategies when using the biLSTM model (**A**) on participants and (**B**) on activities. The mean performance of all participants and activities, respectively, are presented in the last column titled “mean”.

**Figure 4 sensors-23-01577-f004:**
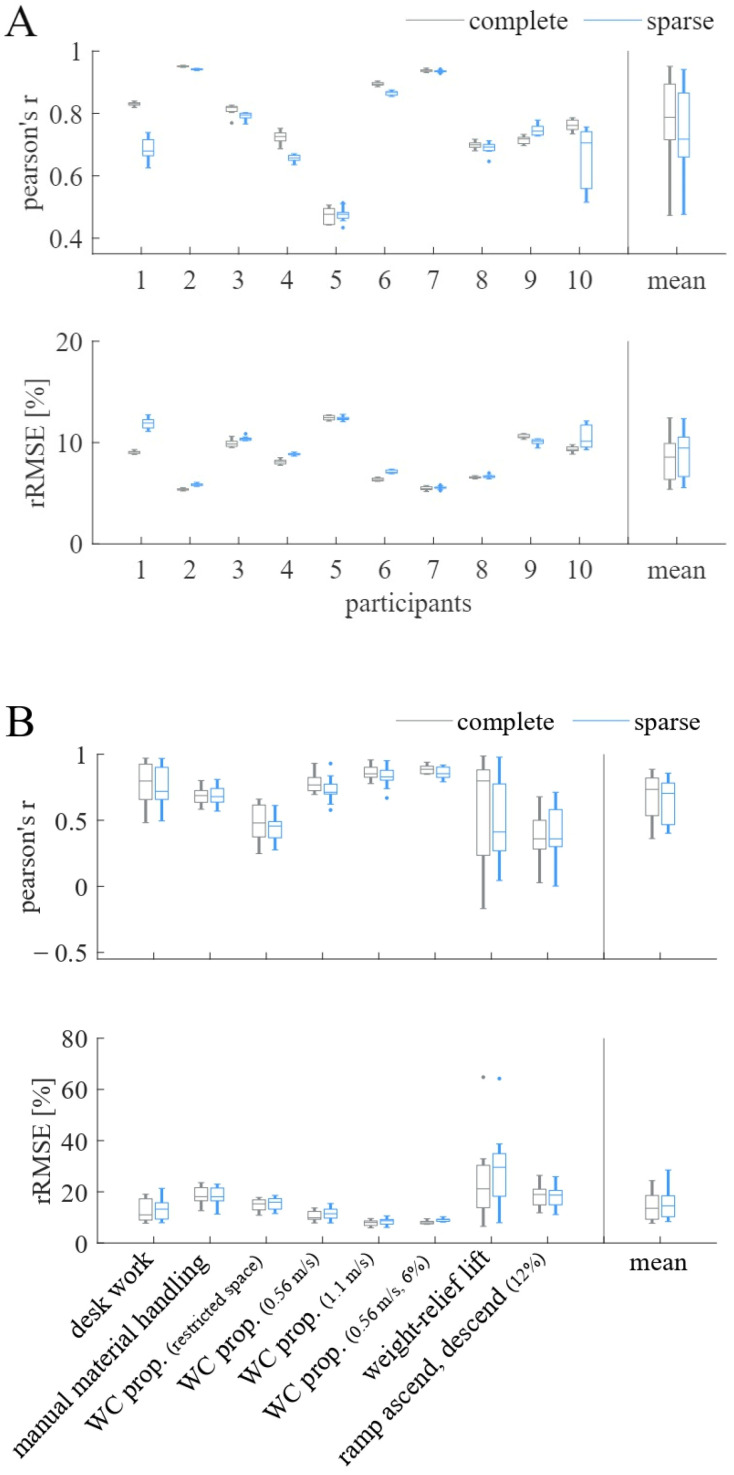
Pearson’s r and relative root-mean-squared error (rRMSE) for comparison of the complete sensor setup and the sparse sensor setup (upper-arm sensor and both WC sensors) when using the LOTO cross-validation strategy and biLSTM model (**A**) on participants and (**B**) on activities. The mean performance of all participants and activities, respectively, are presented in the last column, titled “mean”.

**Figure 5 sensors-23-01577-f005:**
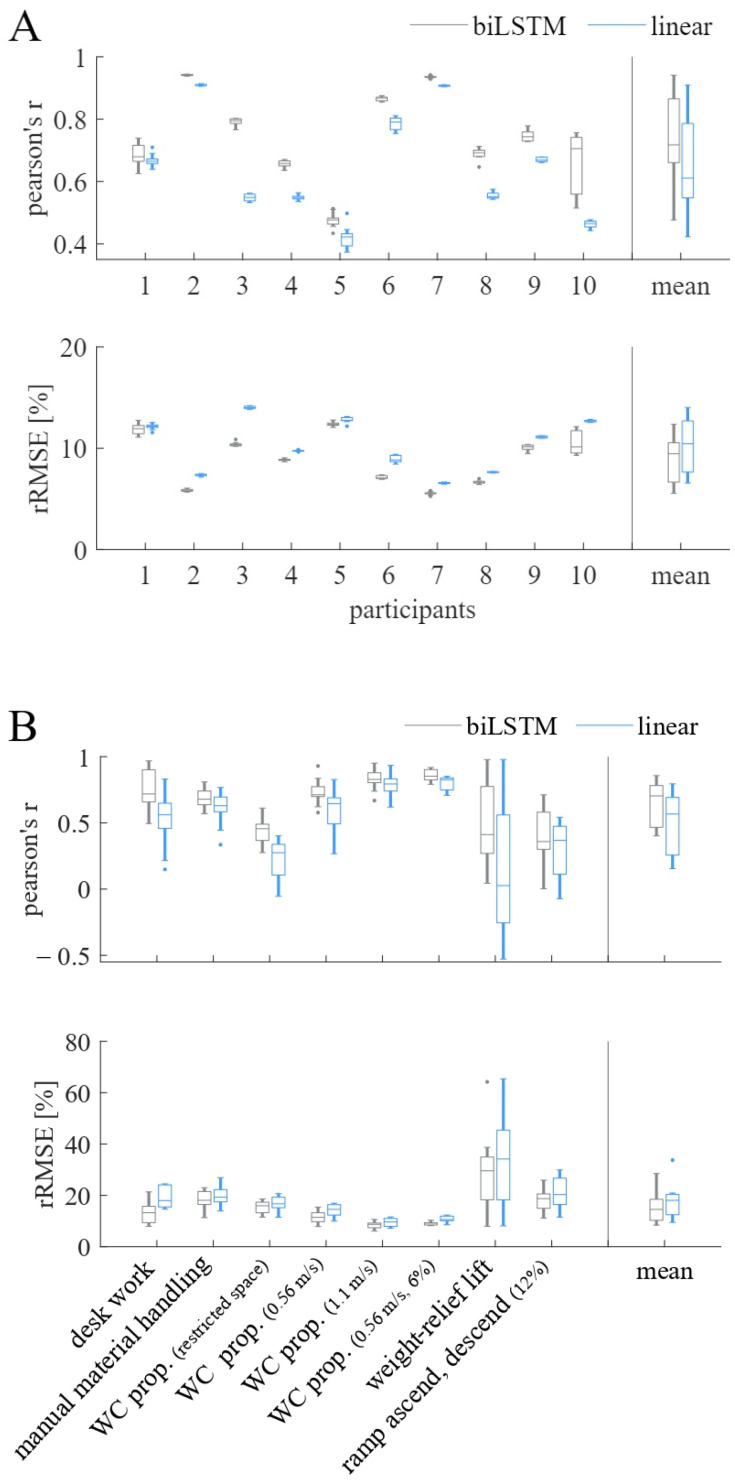
Pearson’s r and relative root-mean-squared error (rRMSE) for comparison of the biLSTM model and the linear model when using the LOTO cross-validation strategy and the sparse sensor setup (**A**) on participants and (**B**) on activities. The mean performance of all participants and activities, respectively, area presented in the last column, titled “mean”.

**Figure 6 sensors-23-01577-f006:**
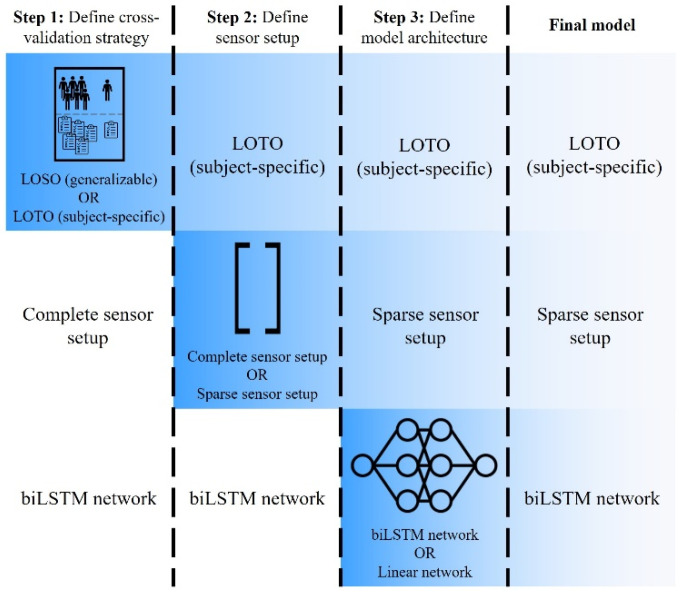
Stepwise approach for finding the best combination of cross-validation strategy (step 1), sensor setup (step 2), and model architecture (step 3) and the respective results.

**Figure 7 sensors-23-01577-f007:**
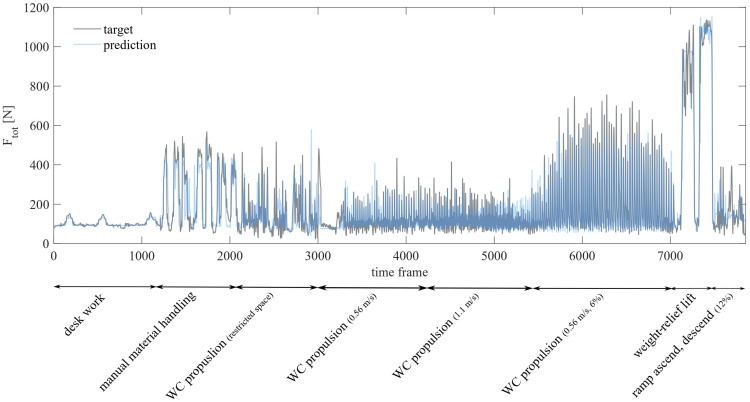
Ground-truth $F_{tot}$ and predicted $F_{tot}$ of one complete iteration for participant 2 using the final model.

**Figure 8 sensors-23-01577-f008:**
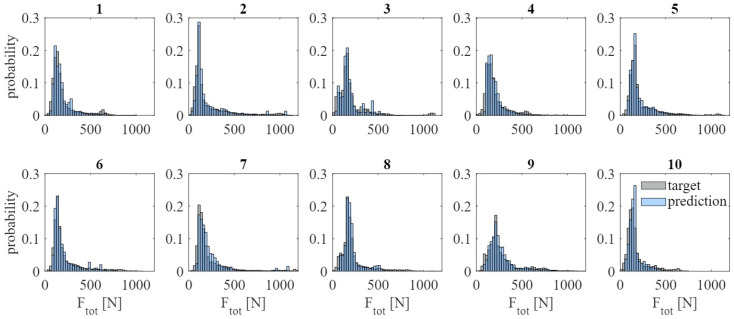
Shoulder-load profiles for each participant over all activities using the final model (bin width = 25 N).

**Figure 9 sensors-23-01577-f009:**
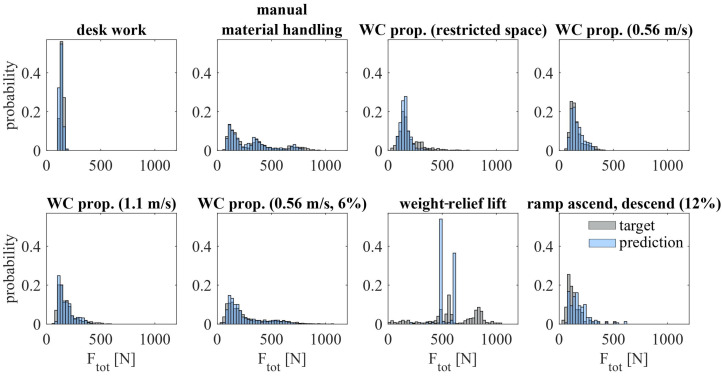
Exemplary shoulder-load profiles for all activities of one participant using the final model (bin width = 25 N).

**Figure 10 sensors-23-01577-f010:**
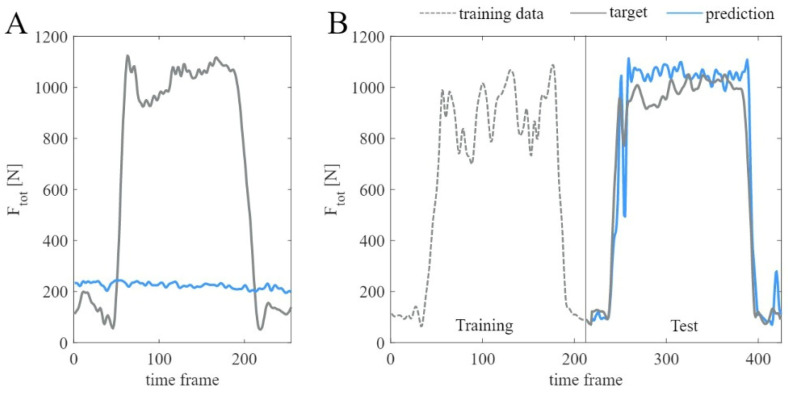
Shoulder joint force for the weight-relief lift activity for participant 5 (**A**) and participant 2 (**B**) using the LOTO cross-validation strategy and biLSTM model with the sparse sensor setup. Activity-related training data for participant 5 were missing from the data set.

**Table 1 sensors-23-01577-t001:** Intersection values for participants, presented as mean (and standard deviation).

Participant	Intersection
1	0.83 (0.01)
2	0.89 (0.00)
3	0.84 (0.01)
4	0.85 (0.00)
5	0.81 (0.01)
6	0.87 (0.01)
7	0.89 (0.00)
8	0.85 (0.01)
9	0.86 (0.01)
10	0.83 (0.01)
Mean	0.85 (0.03)

**Table 2 sensors-23-01577-t002:** Intersection values for activities, presented as mean (and standard deviation).

Activity	Intersection
Desk work	0.97 (0.01)
Manual material handling	0.81 (0.02)
WC propulsion (restricted space)	0.78 (0.03)
WC propulsion (0.56 m/s)	0.92 (0.02)
WC propulsion (1.1 m/s)	0.93 (0.01)
WC propulsion (0.56 m/s, 6%)	0.89 (0.01)
Weight-relief lift	0.66 (0.23)
Ramp ascend, descend	0.77 (0.10)
Mean	0.84 (0.10)

## Data Availability

The data presented in this study are available on request from the corresponding author.

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
