# Peer review of "Machine-Learning-Based Methodology for Estimation of Shoulder Load in Wheelchair-Related Activities Using Wearables"

_sensors, 2023, doi:10.3390/s23031577_

Round 1

Reviewer 1 Report

Authors have written an article related to estimating shoulder load in wheelchair-related activities. I would suggest the following points to authors,

1.       Discuss IMU and EMG data in more detail like the nature of data, size, attributes, etc. There is no sufficient information about dependent and independent variables. Add a separate section to explain detailed information on dataset creation.

2.       What is the novelty of the research work? Elaborate.

3.       I feel the use of biLSTM or ANN does not make much sense in solving the problem statement of the article. Or for the considered problem statement, machine learning is not the solution. It is bit confusing here.

4.       What is the reason for choosing biLSTM? Why not other methods?

5.       Is LOSO and LOTO suitable for time series data?

6.       The article lacks clarity in terms of comparative analysis of biLSTM with other techniques.

7.       The problem statement is not clear.

8.       How exactly recorded wheelchair activities are quantified?

The authors have considered an interesting topic. But it lacks clarity of using biLSTM to address the problem statement. The above points need to be addressed, and paper needs major revision.

Author Response

Please find attached Rebuttal document

Reviewer 2 Report

The study of Amerein and Colleagues aims to develop a data driven model for the estimation of Force load exchange by the shoulder by manual wheelchair users. Overall the study is well written and the goal of the study is clear and deserves investigation. However some concerns should be addressed in order to render the paper available for publication.

CONCERNS

1- The Introduction should report interesting papers regarding the use of EMG for shoulder activity that are not only related to Kinetic data regression but also related to pattern recognition problem in order to present an actual state of the art in the field. For this reason I suggest the Authors to review and report the following papers :

- "Shoulder motion intention detection through myoelectric pattern recognition." IEEE Sensors Letters 5.8 (2021): 1-4.

-Shoulder girdle recognition using electrophysiological and low frequency anatomical contraction signals for prosthesis control." CAAI Transactions on Intelligence Technology 7.1 (2022): 81-94.

- "Real-time classification of shoulder girdle motions for multifunctional prosthetic hand control: A preliminary study." The International Journal of Artificial Organs 42.9 (2019): 508-515.

-"Role of the Window Length for Myoelectric Pattern Recognition in Detecting User Intent of Motion." 2022 IEEE International Symposium on Medical Measurements and Applications (MeMeA). IEEE, 2022.

2) Authors provide LOSO and LOTO validation method and they showed that inter-subjet errors are consistent such that they does not permit generalization of the model across subjects. This is can be due to the presence of EMG signals which naturally carry inter-subjecct generalization of both regression and pattern recognition problems. Authors should try to develop model using only IMU data, hence observing whether LOSO errors reduce. This can help to understand which kind of data (IMU, EMG or IMU+EMG) can provide the best framework for obtaining a generalized inter-subject model.

Author Response

Please find attached rebuttal document.

Round 2

Reviewer 1 Report

Though the authors have addressed a few points, points no.  3, 4, and 6 of the first review remain unanswered. Please address them and submit the revised article.

Author Response

Please find attached rebuttal letter dd 2023.01.25

Reviewer 2 Report

Authors provided good explanations behind the methodological points. Thus, the paper can be accepted.

Author Response

(The authors gave the same response as above.)
